# Pre-Sleep Casein Supplementation, Metabolism, and Appetite: A Systematic Review

**DOI:** 10.3390/nu13061872

**Published:** 2021-05-30

**Authors:** Justin Dela Cruz, David Kahan

**Affiliations:** School of Exercise and Nutritional Sciences, San Diego State University, San Diego, CA 92182, USA; dkahan@sdsu.edu

**Keywords:** casein, sleep, metabolism, appetite, exercise

## Abstract

Protein intake is an important factor for augmenting the response to resistance training in healthy individuals. Although food intake can help with anabolism during the day, the period of time during sleep is typically characterized by catabolism and other metabolic shifts. Research on the application of nighttime casein protein supplementation has introduced a new research paradigm related to protein timing. Pre-sleep casein supplementation has been attributed to improved adaptive response by skeletal muscle to resistance training through increases in muscle protein synthesis, muscle mass, and strength. However, it remains unclear what the effect of this nutritional strategy is on non-muscular parameters such as metabolism and appetite in both healthy and unhealthy populations. The purpose of this systematic review is to understand the effects of pre-sleep casein protein on energy expenditure, lipolysis, appetite, and food intake in both healthy and overweight or obese individuals. A systematic review following PRISMA guidelines was conducted in CINAHL, Cochrane, and SPORTDiscus during March 2021, and 11 studies met the inclusion criteria. A summary of the main findings shows limited to no effects on metabolism or appetite when ingesting 24–48 g of casein 30 min before sleep, but data are limited, and future research is needed to clarify the relationships observed.

## 1. Introduction

Timing of food intake is a major component influencing individual total daily intake. Food consumption primarily in the earlier part of the day is correlated with decreased overall daily intake, whereas consumption closer to nighttime can increase overall daily intake [1]. This relationship is affected in part by macronutrient intake, as consumption of either carbohydrates, fats, or proteins in the morning all have a much higher rate of satiation compared to nighttime consumption [2]. With such prominent effects of timing on total daily food intake, the overconsumption of food in the latter half of the day can be linked to excessive weight gain. Nighttime eating behaviors are a contributing factor to additional yearly weight gain in obese women [3]. This relationship may be linked to physiological changes dictated by the circadian rhythm that cause insulin sensitivity and energy expenditure to decrease in the latter half of the day compared to in the morning [4,5]. Furthermore, this issue may be exacerbated, as chronic nighttime eating behaviors can also develop into disordered eating patterns such as nighttime eating syndrome (NES) or nocturnal eating/drinking syndrome (NEDS) [6].

Although weight gain may be linked to increased food intake during nighttime eating behaviors, this may also be affected by metabolic adaptations that occur during sleep. Metabolic processes are altered during times of sleep through a reduction in both energy expenditure and fat oxidation [7]. Nighttime eating can also negatively alter metabolism by shifting fuel utilization away from fat and toward carbohydrates [8]. However, the detrimental effects of nighttime feeding that come with the classic consumption of large calorie-dense meals with mixed macronutrient composition can potentially be mitigated by the consumption of small nutrient-dense and low-calorie meals that focus on one specific macronutrient [9]. 

Recent research has delved into the beneficial effects of nighttime single macronutrient consumption, specifically protein intake in close proximity to sleep, and its effects on muscular adaptations from resistance training. It is important to note that the consumption of protein significantly decreases the rate of total body protein breakdown [10], which supports the theoretical framework for consuming protein around a time of catabolism such as sleep. One of the first research studies to look at pre-sleep intake of casein protein reported the enhancement of protein synthesis rates as a skeletal muscle mechanism for recovery after resistance training [11]. This was later confirmed in one other study [12], although another study later differentiated that muscle connective tissue protein synthesis rate was unaffected [13]. Another study found that pre-sleep casein supplementation augments protein synthesis rates via the mechanism of increasing the incorporation of amino acids [14]. These findings may explain the underlying mechanisms as to why one study done on pre-sleep casein supplementation was shown to increase muscle cross-sectional area, type 2 fiber size, and strength in healthy young men during a 12-week resistance training program [15]. Even with what is known about altered metabolism during sleep and the potential risks associated with nighttime eating on weight gain, pre-sleep casein supplementation does not increase fat mass in a healthy active population [16]. This nutritional strategy presents an advantageous perspective for increasing protein intake during the nighttime in a healthy active population without any detriments to body composition. 

A recent systematic review looked at the effects of pre-sleep casein supplementation on muscle adaptations in healthy individuals and reported significant increases in whole body protein synthesis observed in five studies [17]. However, this systematic review had limited evidence regarding other muscle adaptations, with only one included study reporting significant findings in muscle cross-sectional area, type 2 fiber size, and strength [17]. Furthermore, there have been no reviews published regarding the impact of this nutritional strategy on non-muscular parameters such as metabolism and appetite. Dairy foods such as milk, cheese, or yogurt have shown significant effects on reducing appetite [18]. Casein has been reported to decrease energy intake to a greater extent than whey protein [19]. Although casein consumption has been shown to suppress appetite in overweight individuals through elevation of ghrelin and CCK [20], there is a lack of investigation specifically into the impact of pre-sleep casein supplementation on overweight or obese individuals. The aim of this systematic review is to synthesize research findings on the effect of pre-sleep casein supplementation on metabolism and appetite in both healthy and unhealthy populations. 

## 2. Materials and Methods

### 2.1. Systematic Search Strategy

The systematic search for this study followed PRISMA (Preferred Reporting Items for Systematic Reviews and Meta-Analyses) guidelines. Table 1 presents the Population, Intervention, Comparison, Outcome, and Setting (PICOS) parameters that were used to define the scope of research included. 

A systematic search was conducted in March 2021 by one independent author (J.D.C.) and used the following three online databases: CINAHL, Cochrane, and SPORTDiscus. In order to identify all relevant articles published on the topic, no filters were applied. 

Titles/abstracts/keywords were identified using the following Boolean search operators: (“casein” OR “protein”) AND (“sleep” OR “overnight” OR “pre-sleep” OR “presleep” OR “prior to sleep*” OR “before sleep*”) AND (“metabol*” OR “energy expenditure” OR “caloric expenditure” OR “basal metabolic rate” OR “BMR” OR “metabolic rate” OR “resting energy expenditure” OR “REE” OR “resting metabolic rate” OR “RMR” OR “lipolysis” OR “fat metabolism” OR “fat oxidation” OR “beta oxidation” OR “β oxidation” OR “fat breakdown” OR “hunger” OR “appetite” OR “satiety” OR “energy intake” OR “food intake” OR “food consumption”).

### 2.2. Eligibility Criteria

Articles were included in this systematic review if they met the following criteria: (1) randomized control trial; (2) sample of adult/elderly individuals (≥18 years old); (3) casein protein was consumed pre-sleep; (4) casein supplementation was compared to a control treatment; and (5) assessed the effect of pre-sleep casein supplementation on metabolism or appetite. The exclusion criteria were (1) no pre-sleep casein supplementation consumed; (2) no metabolic or appetite parameters assessed; and (3) abstracts, reviews, manuscripts, thesis, books, letters, conference posters, or public seminars.

Casein intake was considered as pre-sleep if it was the last meal consumed during that day. Metabolism was used as an umbrella term and operationally defined as (1) any form of energy expenditure or metabolic rate, or (2) lipolysis or fat oxidation. Appetite was operationally defined as (1) appetite or satiety, (2) hunger or desire to eat, or (3) food intake. A narrative synthesis of the results from the selected studies included in this review was classified into the metabolic and appetite parameters previously defined.

## 3. Results

### 3.1. Systematic Search and Study Selection

The systematic search and article selection process is depicted in Figure 1. Preliminary searching across the three databases identified a total of 1964 articles that were exported into the electronic software program Zotero for literature management. After duplicates were removed, 1904 articles remained, and all articles were published between 1978 and 2021, but we did not take into account articles from February to March 2021. Articles identified were assessed using the inclusion criteria, which condensed the search down to 44 articles that were exported into the online electronic program Covidence for systematic review management. During the screening process, a secondary round of deduplication (n = 23) and evaluation of article title/abstract that did not meet inclusion criteria (n = 8) was completed. A total of 13 articles remained for full-text evaluation for eligibility. 

Full-text evaluation for eligibility excluded four articles for either improper publication format (manuscript or conference poster; n = 3) or if the study did not measure any metabolic or appetite parameters (n = 1). Additional studies retrieved from the reference section of the 13 original articles were also screened for eligibility and added to the systematic review (n = 2). A total of 11 articles were included in this systematic review for data extraction and synthesis of the main results.

### 3.2. Descriptive Data and Characteristics of Selected Studies

The main characteristics of the selected studies in this systematic review are recorded in Table 2. All studies were randomized control trials, and 10 of the 11 studies were blinded (nine double blind; one single blind). Five out of the 11 studies included a crossover design. Casein consumed in the selected studies ranged between 24 and 48 g doses. Eight out of the 11 studies used a casein protein drink as the food medium for pre-sleep supplementation, and the remaining three studies used either cottage cheese or a casein supplement mixture (casein with whey and carbohydrate or casein with tryptophan). 

Table 3 lists the descriptive details of the studies that included an exercise trial or meal standardization. Six out of the 11 studies included a standardized single meal or entire day meal plan based on the participants’ estimated energy expenditure. Only four out of the 11 studies included an exercise trial as part of the research protocol.

Six out of the 11 studies observed metabolic or appetite parameters in healthy young adult samples, four studies were conducted in overweight or obese young adult samples, and one study was conducted in a healthy elderly adult sample. In order to differentiate the main findings, Table 4 categorized main results into metabolic and appetite parameters measured (i.e., next-day appetite/hunger/satiety/food intake, energy expenditure/metabolic rate, and lipolysis/fat oxidation) and were further subdivided into sample population (i.e., healthy young adults, overweight or obese young adults, or healthy elderly adults).

### 3.3. Effects of Pre-Sleep Casein Supplementation on Next-Day Appetite, Hunger, Satiety, and Next-Day Food Intake

Results of studies that observed the effect of casein on appetite sensations were notably influenced by the population involved. All studies in healthy and active young adults or active elderly adults (n = 7) found no significant effect on appetite sensations (*p* > 0.05). However, studies involving overweight or obese individuals noted conflicting findings.

Three of the four studies on overweight or obese individuals observed improvements in appetite sensations. Ormsbee et al. [23] noted that 30 g of pre-sleep casein consumption for 4 weeks resulted in a significant increase in morning satiety compared to 30 g of whey protein or 34 g of maltodextrin (*p* = 0.02). Kinsey et al. [21] also found a significant increase in morning satiety (*p* = 0.03) and a decrease in the desire to eat (*p* = 0.006) for 30 g of casein compared to 30 g of whey protein or 34 g of maltodextrin. Lay et al. [25] noted a greater increase on next-morning fullness for pre-sleep casein consumption compared to a non-nutritive control, but their data were not statistically significant (*p* = 0.07). It should be noted that Lay et al. did not use a standard casein protein formulation, but rather 2 skim milk powder mixtures that also included whey and carbohydrates (BS10= 8 g casein, 2 g whey, 14 g carbohydrates; BS30 = 24 g casein, 6 g whey, 42 g carbohydrates). Only the study by Kinsey et al. [24] found an opposite effect on appetite, with 30 g of casein causing a significant increase in desire to eat (*p* = 0.03).

### 3.4. Effects of Pre-Sleep Casein Supplementation on Energy Expenditure and Metabolic Rate

None of the studies in healthy young adults or active elderly adults (n = 6) observed a significant difference (*p* > 0.05) in metabolism, except for one. Madzima et al. [22] found that when comparing two casein formulations (24 g or 48 g) compared to two whey protein formulations (24 g or 48 g) or one non-nutritive control, a significant change in VO2 was seen for the group that consumed casein. 

In the studies with overweight or obese participants, no significant differences were found in metabolism either, but Ormsbee et al. [23] found a non-statistically significant (*p* = 0.07) increase in resting metabolic rate (RMR) in their casein group compared to their control group (see Table 4. Summary of main results from included studies related to pre-sleep casein supplementation effects on metabolism and appetite).

### 3.5. Effects of Pre-Sleep Casein Supplementation on Lipolysis and Fat Oxidation

Only three of the 11 studies observed the effects of pre-sleep casein supplementation on lipolysis in either healthy young adults or overweight and obese young adults. 

Two of the three studies used 30 g of casein for their experimental trial and found no significant differences (*p* > 0.05) in fat oxidation. These studies measured lipolysis in subcutaneous abdominal adipose tissue (SCAAT) directly through microdialysis, which monitors interstitial glycerol concentrations over an extended period of time.

Madzima et al. [27] found a significant decrease (*p* = 0.04) in fat oxidation in healthy young adults when comparing their 24 g casein group to their 24 g whey protein group, but there was no significant difference between the casein group compared to the placebo group. Their protocol measured fat oxidation indirectly through respiratory exchange ratio from the same metabolic cart and a ventilated hood setup used for indirect calorimetry.

## 4. Discussion

This systematic review investigated the effects of pre-sleep casein supplementation on metabolic and appetite parameters. A summary of the main findings indicates that ingesting 24–48 g of casein 30 min before sleep does not suppress next-day appetite sensations and food intake in healthy young adults. Effects on appetite sensations and food intake in overweight or obese populations are unclear, as findings are conflicting, and a definitive relationship cannot be stated on whether pre-sleep casein supplementation increases or decreases next-day appetite and food intake. No effects on metabolic parameters were observed in overweight or obese populations or healthy elderly adults. Although there are some beneficial findings of pre-sleep casein supplementation on energy expenditure and lipolysis in healthy young adults, findings are limited, and more studies will be needed to elucidate these results. Furthermore, only one selected study in this systematic review included a healthy adult elderly population, limiting the representation of these findings to a fairly small sample size. These results support the efficacy of pre-sleep casein supplementation as a means to enhance muscular adaptations in a healthy active population without causing unwanted metabolic responses. However, further research is needed to clarify the potential utilization of this strategy for limiting weight gain in overweight or obese individuals.

This is relatively important for individuals trying to maximize muscle mass gain but should not be considered the most optimal strategy. It is important to know that protein distribution spread evenly throughout the day is associated with higher muscle mass [32]. One study [33] showed that evenly distributed protein intake in healthy adults significantly increased total daily mixed muscle protein fractional synthesis rate by up to 25% compared to an uneven protein intake distribution. Therefore, pre-sleep casein supplementation serves as an additional feeding window for extending protein distribution evenly throughout the day, theoretically creating a longer period of elevated muscle protein synthesis (MPS) for muscular adaptation. However, one study [34] noted that optimizing protein distribution throughout the day is less important compared to optimizing total daily protein intake primarily. Taking this into consideration, individuals trying to maximize muscular adaptation to resistance training should focus foremost on meeting adequate total protein intake of 0.8–1.3 g/kg/day [34] before considering this pre-sleep protein supplementation strategy.

Furthermore, pre-sleep casein supplementation might not be the best protein choice for meeting general protein requirements in all healthy individuals trying to maximize adaptations to resistance training. Although casein elicits a similar MPS response regardless of age in a healthy population [35], whey protein typically stimulates a greater MPS response at rest and after resistance exercise [36]. This in part is due to whey’s ability to elevate plasma amino acid concentrations to a greater extent than casein [36,37]. In addition, whey protein is characterized by faster digestion, better absorption kinetics, and greater total leucine content [38]. These gastrointestinal and protein characteristics are advantageous for maximizing short-term MPS response, since whey stimulates a greater peak MPS elevation within the first 1 to 3 h after consumption before gradually declining [39]. During the day, food is readily available and can be consumed around the same time MPS starts to decrease, so that maximal MPS response remains constant for long periods of time. In comparison, casein elicits slower gastrointestinal responses via gastric emptying, which is up to two times slower than whey protein [40]. Although the peak elevation of MPS response in casein is lower than whey within 1 to 3 h after food intake, casein can sustain a greater average peak MPS response over a 6 h time frame after initial food intake [39]. These influencing determinants reinforce casein as the optimal protein choice for consumption during nighttime feeding, since sleep creates a restrictive time frame where access to feeding for maximizing MPS is not feasible.

The findings of this systematic review also provide a differing perspective on nighttime snacking or eating, which has previously shown detrimental effects depending on the food being consumed. One study observed a significant decrease in fat oxidation but no significant changes in energy intake, energy expenditure, or body weight when food consumed was primarily carbohydrate and fat (mean protein/fat/carbohydrate ratio composition of a ≈200 kcal snack was 5:50:45) [41]. The findings of this systematic review further support that some of the detrimental effects of nighttime feeding can be negated with single macronutrient consumption, specifically protein. Pre-sleep casein supplementation can be implemented as an adjunct strategy for obese or overweight individuals to meet the entirety of a high-protein diet, which shows beneficial effects on all parameters regarding whole-day “satiety, thermogenesis, sleeping metabolic rate, protein balance, and fat oxidation” [42]. The beneficial effects on appetite and food intake even occur when hormonal responses such as decreased leptin and increased ghrelin occur [43]. Beneficial effects of high protein intake are not exclusive to total daily intake, as these effects have even been seen in single high-protein meals as well [44]. Milk proteins are one of the most optimal protein sources when trying to blunt hunger, as they improve satiety through the modulation of gastric emptying [40]. Specific to the timing of casein intake, it may also be a more beneficial nighttime snack option than whey protein because of its greater satiety effects [45]. Even at breakfast, high-casein meals display beneficial effects on appetite [46]. With these considerations, overweight or obese individuals can structure high-protein meal intake around times when the strongest appetite sensations occur. Therefore, pre-sleep casein supplementation may be an effective strategy for overweight or obese individuals who typically experience greater sensations of hunger at night.

Pre-sleep casein supplementation also shows potential benefits for meeting protein requirements in elderly individuals. The elderly population is at heightened risk of sarcopenia, which can limit mobility [47] and can also elevate risk of mortality [48]. It is important to understand that aging introduces additional factors that can negatively affect the efficacy of resistance training for maintaining lean mass and strength. Aging is associated with a weaker MPS response to dietary protein intake [49] and a decrease in both appetite and energy intake [50]. These physiological changes are important to understand in the context of optimal protein dosage for stimulating maximal MPS, as higher intakes are required in older adults compared to younger adults [49]. Supplementing daily intake in this population with nighttime protein feeding presents a possible solution to the physiological obstacles that elderly individuals face.

The main findings from this systematic review support the efficacy of pre-sleep casein supplementation, but limitations of the included studies should still be taken into consideration. Only half of the studies had meal standardization, which may have evoked undetected effects on the results obtained in the studies that did not include such standardization. Furthermore, the application of pre-sleep casein supplementation in research is typically focused around beneficial adaptations during resistance training, but fewer than half of the studies included an exercise component within their protocol. This is an important consideration, as higher doses of up to 40 g compared to 20–30 g of casein have clearly been shown to increase utilization within muscle protein synthesis overnight and to an even greater extent when combined with resistance training [51]. Another limitation to take into consideration is the food source that provided casein. Most studies used a liquid formulation of micellar casein with only a few studies providing casein in a mixed formulation with other macronutrients or as cottage cheese. The consumption of casein alone typically has a lower effect on appetite regulation than actual milk [52]. Future research should use a broader sample population to address these limitations. Only one study was in healthy elderly individuals, and no study was done in an overweight or obese elderly population. Lastly, only one study looked at the chronic effect of casein supplementation over the span of several weeks compared to the acute effects observed in most of the studies. Future research should be directed toward addressing these limitations through the incorporation of meal standardization and resistance training protocols, consumption of casein through different food sources, greater inclusion of healthy and unhealthy elderly adults, and analysis of chronic pre-sleep casein supplementation.

## 5. Conclusions

The findings of this systematic review show an overall trend of limited to no effect of pre-sleep casein supplementation on metabolic or appetite parameters in healthy and unhealthy populations. All findings from this systematic review should be interpreted with caution, as data are limited, and future research is needed to clarify the relationships observed. 

## Figures and Tables

**Figure 1 nutrients-13-01872-f001:**
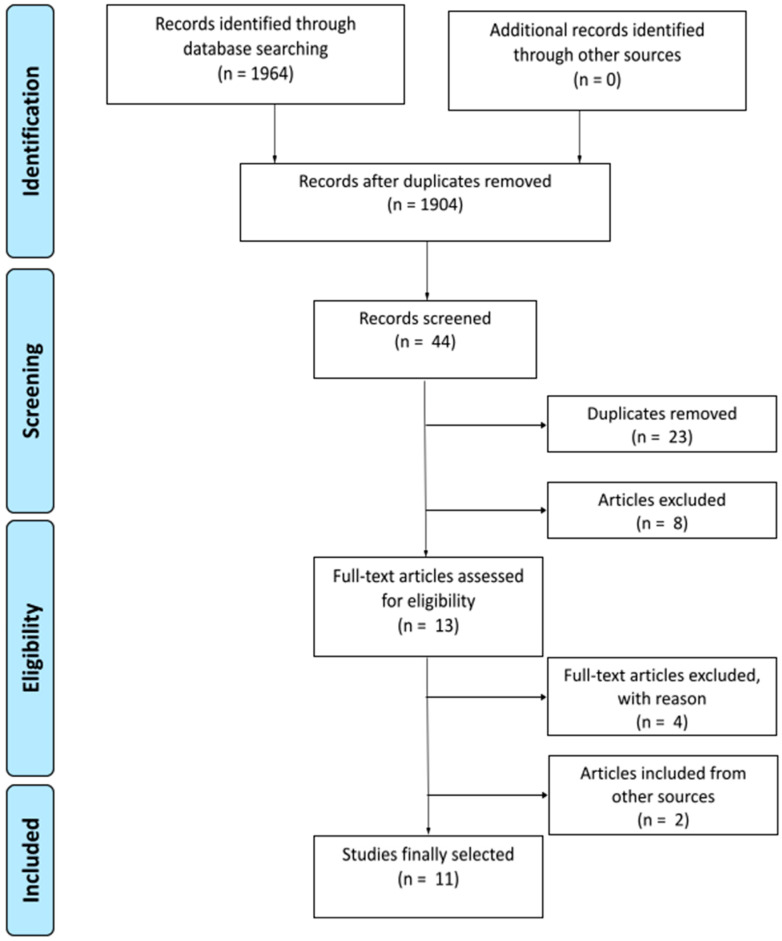
PRISMA flow diagram for study selection process of pre-sleep casein supplementation articles.

**Table 1 nutrients-13-01872-t001:** PICOS (Population, Intervention, Comparison, Outcome, Setting) criteria for inclusion of studies.

Parameter	Description
Population	Human trials in adults or elderly (≥18 years old)
Intervention	Casein supplementation before sleep
Comparison	Casein vs. control (placebo, carbohydrate, or water)
Outcome	Metabolism or appetite
Setting (Study Design)	Randomized control trial

**Table 2 nutrients-13-01872-t002:** Main characteristics of included studies related to pre-sleep casein supplementation effects on metabolism and appetite.

Study (Author, Year)	Subjects	Sample Size (Mean Age)	Study Design	Metabolic Parameters Measured	Protein Source	Exercise Trial Included?		Standardized Meal?
Kinsey et al., 2014 [21]	Obese and overweight women	44(29)	DRCT	-Appetite -Metabolism (RMR)	Casein	No		No
Madzima et al., 2014 [22]	Active men	11(24)	DRCT, CD	-Hunger, satiety, desire to eat -Metabolism (REE)	Casein	No		No
Ormsbee et al., 2015 [23]	Obese women	37(29)	DRCT	-Appetite -Metabolism	Casein	Yes		No
Kinsey et al., 2016 [24]	Obese men	12(27)	DRCT	-Appetite -Metabolism -SCAAT lipolysis	Casein	No		Yes
Lay et al., 2018 [25]	Overweight men	8(24)	DRCT, CD	-Hunger, fullness, desire to eat -Next day ad libitum breakfast	Casein mixture with whey and carbohydrate	No		Yes
Leyh et al., 2018 [26]	Active women	10(23)	DRCT, CD	-Appetite -Metabolism (REE)	Casein and cottage cheese	No		Yes
Madzima et al., 2018 [27]	Active women	9(25)	DRCT, CD	-Metabolism	Casein	Yes		No
Trommelen et al., 2018 [28]	Active men	36(23)	DRCT	-Next morning hunger and satiety	Casein		Yes	Yes
Allman et al., 2020 [29]	Active men	13(22)	DRCT, CD	-Metabolism (REE) -SCAAT lipolysis	Casein		Yes	Yes
Morehen et al., 2020 [30]	Older men and women	12(71)	SRCT	-Appetite -Metabolism -Next day ad libitum breakfast	Casein		No	Yes
Nelson et al., 2021 [31]	Active women	13(23)	RCT	-Appetite -Metabolism	Casein mixture with tryptophan		No	No

Abbreviations: double-blind randomized control trial (DRCT), single-blind randomized control trial (SRCT), randomized control trial (RCT), crossover design (CD), subcutaneous abdominal adipose tissue (SCAAT), resting metabolic rate (RMR), resting energy expenditure (REE).

**Table 3 nutrients-13-01872-t003:** Descriptive characteristics of studies that included meal standardization or resistance exercise.

Study (Author, Year)	Exercise Trial			Meal Standardization	
	Exercise Modality	Exercise Trial Length	Exercise Protocol	Single Meal or Entire Day	Energy Expenditure Estimate
Ormsbee et al., 2015 [23]	Resistance training and HIIT	4 week (3 days/week nonconsecutively) 2 days of resistance training and 1 day of HIIT	Resistance training -Total of 3 sets per exercise (first 2 sets for 10 repetitions and last set performed to muscular exhaustion). -Exercises performed included: chest press, seated row, leg press, shoulder press, leg extension, and leg curl. -Chest press and leg press at 70–85% of 1RM and the remaining exercises at a weight that could be lifted for 10–12 repetitions. -Total of 90–120 s rest periods. HIIT -Based on individual RPE scale (1–10). -Self-selected machine (cycle ergometer, treadmill, or elliptical trainer). -Total of 4 HIIT cycles performed for a total of 20 min. -HIIT cycle: warm up at RPE 5 (2 min), increased by 1 RPE every minute until reaching RPE 9, RPE reduced to 6 for 1 min, repeated ramping cycle up to RPE 9.		
Lay et al., 2018 [25]				Entire day	-Distribution of 50% CHO, 32% FAT, 18% PRO -Energy matched to the participants’ average evening meal intake from their food record.
Leyh et al., 2018 [26]				Single meal	Energy expenditure estimate and macronutrient distribution details not stated.
Madzima et al., 2018 [27]	Resistance training	Single day session	-Exercises performed: chest press, leg press, lat pull-down, shoulder press, leg extension, and leg curl. -Performed at metronome cadence of 30 beats/minute (ratio of 2:2 s concentric and eccentric). -Total of 3 sets per exercise (first two sets for 10 repetitions and last set performed to muscular exhaustion). -Exercises performed at 60% of 1RM.		
Allman et al., 2020 [29]	Resistance training	Single day session	-Exercises performed: back squat, bench press, Romanian deadlift, bent-over row, shoulder press, and reverse lunges. -Total of 4 sets of 10 repetitions (Set 1: 40% of 1RM, Set 2–4: 65% at 65% 1RM).	Entire day	-Distribution of 40% CHO, 30% FAT, %30 PRO -Energy matched to the participants’ Cunningham equation calculation.
Morehen et al., 2020 [30]				Single meal	-Distribution of 50% CHO, 32% FAT,18% PRO -Energy matched to the participants’ habitual intake record.

Abbreviations: high-intensity interval training (HIIT), one-repetition maximum (1RM), rated perceived exertion (RPE), carbohydrate (CHO), protein (PRO).

**Table 4 nutrients-13-01872-t004:** Summary of main results from included studies related to pre-sleep casein supplementation effects on metabolism and appetite.

Study (Author, Year)	Next-Morning Appetite, Hunger, and Satiety	Metabolism	Lipolysis	Next-Morning Food Intake
Healthy young adult				
Madzima et al., 2014 [22]	No effect on appetite sensations (*p* > 0.05).	No group x time interaction for RMR (*p* > 0.05). Mean VO2 significantly greater in casein than in control (*p* < 0.0001).		
Leyh et al., 2018 [26]	No effect on appetite sensations (*p* > 0.05).	No effect on metabolism (*p* > 0.05).		
Madzima et al., 2018 [27]		No effect on metabolism (*p* > 0.05)	24 g casein had significantly lower (*p* = 0.04) fat oxidation compared to 24 g whey when measured indirectly by RER.	
Trommelen et al., 2018 [28]	No effect on appetite sensations (*p* > 0.05).			No effect on next-morning food intake (*p* > 0.05).
Allman et al., 2020 [29]		No effect on metabolism (*p* > 0.05).	No effect on lipolysis (*p* > 0.05).	
Nelson et al., 2021 [31]	No effect on appetite sensations (*p* > 0.05).	No effect on metabolism (*p* > 0.05).		
Overweight/obese young adult				
Kinsey et al., 2014 [21]	No group x time interaction for any appetite sensations (*p* > 0.05). Significant main effect of time interaction: Increased satiety (*p* = 0.03) Reduced desire to eat (*p* = 0.006)	No time or group x time interaction for RMR (*p* > 0.05).		
Ormsbee et al., 2015 [23]	No effect on hunger or desire to eat (*p* > 0.05). Significant group x time interaction: Casein increased morning satiety compared to controls after 4 weeks (*p* = 0.02).	No group x time interaction for RMR (*p* > 0.05). Not statistically significant (*p* = 0.07) but casein had greater increases on RMR than in control.		
Kinsey et al., 2016 [24]	No effect on hunger or satiety (*p* > 0.05). Significant group effect: Casein increased desire to eat compared to control (*p* = 0.03).	No effect on metabolism (*p* > 0.05).	No effect on lipolysis (*p* > 0.05).	
Overweight/obese young adult				
Lay et al., 2018 [25]	No effect on hunger or desire to eat (*p* > 0.05). Not statistically significant (*p* = 0.07) but casein had greater increases on next-morning fullness.	No effect on metabolism (*p* > 0.05).		No effect on next-morning food intake (*p* > 0.05).
Healthy elderly adult				
Morehen et al., 2020 [30]	No effect on appetite sensations (*p* > 0.05).	No effect on metabolism (*p* > 0.05).		No effect on next-morning food intake (*p* > 0.05).

## Data Availability

Data sharing not applicable.

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
