# Peer review of "Pre-Sleep Casein Supplementation, Metabolism, and Appetite: A Systematic Review"

_nutrients, 2021, doi:10.3390/nu13061872_

Round 1
Reviewer 1 Report
Title: Pre-Sleep Casein Supplementation, Metabolism, and Appetite: A Systematic Review
Authors: Justin Dela Cruz 1 and David Kahan
Summary: Authors conducted systematic review following PRISMA guidelines and found no detrimental effects on metabolism or appetite when ingesting 24-48 grams of casein 30 minutes before sleep. This review adds to the body of knowledge on the effects of pre sleep casein supplementationn on non-muscle outcomes, such as appetite, hunger and next day food intake, which may be of some importance for the field.
Major revisions:
1. Currently, there are no placebo-controlled RCTs that show significant effects of pre sleep protein supplementation on muscle mass or strength (especially not using protein-based placebos such as collagen). As such, authors must remove or soften all statements made on the effects of casein supplementation on muscle mass and strength, specifically, in the abstract, introduction and discussion. Of note, reference 15 (Joy M et al) is a preliminary study showing no difference between day time vs night time protein supplementation on mass and strength in resistance trained individuals. In addition, the study did not employ a resistance trained group alone or an appropriate placebo group. Further to this, the cited systematic review (reference 17) clearly states that the current available evidence on mass and strength must be interpreted with caution. Lastly, Luc van Loon's groups have clearly shown that the overall evidence only supports the intake of 40 g of casein together with resistance exercise in the evening for an increase in muscle protein synthesis (MPS) over placebo. In other words, there is a dose effect and 20 g of casein is simply not enough for enhancing MPS in either young or old adults. Please reword and adjust your paper accordingly.
2. Author must soften and clarify their findings in the abstract, discussion and conclusion to not (non-purposely) mislead the readers. Specifically, their evidence indicate that night-time casein supplementation does not suppress appetite, hunger or next day food intake in healthy adults, and that more more data is needed to elucidate the trending effects in overweight and obese subjects and if pre-sleep protein supplementation may be a useful strategy to limit weight gain in these individuals.
As a whole, pre-sleep protein supplementation does have merit to enhance MPS, but more research is needed to elucidate what kind of protein source is optimal, dosages and population differences. This study may add to the body of knowledge regarding non-muscle outcomes, such as appetite, hunger and next day food intake, which is of some importance for the field as well.
Author Response
- All statements regarding systematic review findings were reworded in the abstract, discussion, and newly added conclusion section. The Joy et al. reference was removed and replaced by Snijders et al. (2015), which is a study that observed increased muscle cross sectional area, type 2 muscle fiber, and strength with pre-sleep casein supplementation on young men during 12-week resistance training protocol. The reference (17) regarding the systematic review on pre-sleep protein consumption on muscle-related outcomes discussed in the introduction has now been reworded. Lastly, addition of casein supplementation dosage-response and relationship with resistance training for augmenting such response has been noted in the last paragraph of the discussion section.
- Statements of findings have been softened and reworded.

Reviewer 2 Report
This article was very well written in terms of the methodology of systematic review.
A few minor suggestions for authors
1.The introduction lacks a description of the mechanism (biochemistry) of casein's influence on muscle metabolism and its regeneration in the context of appetite
2.the submitted studies did not take into account the articles from February-March 2021, e.g.
Holwerda AM, Trommelen J, Kouw IWK, Senden JM, Goessens JPB, van Kranenburg J, Gijsen AP, Verdijk LB, van Loon LJC. Exercise Plus Presleep Protein Ingestion Increases Overnight Muscle Connective Tissue Protein Synthesis Rates in Healthy Older Men. Int J Sport Nutr Exerc Metab. 2021 Feb 14:1-10. doi: 10.1123/ijsnem.2020-0222. Online ahead of print.PMID: 33588378
Costa JV, Michel JM, Madzima TA The Acute Effects of a Relative Dose of Pre-Sleep Protein on Recovery Following Evening Resistance Exercise in Active Young Men.. Sports (Basel). 2021 Mar 26;9(4):44. doi: 10.3390/sports9040044.PMID: 33810526
3. Please add a short conclusion summarizing the study
Author Response
- Description of biochemical mechanism underlying casein's influence on muscle metabolism was not included in the introduction (section 1) for the following reasons: (1) to highlight the main focus of its influence specifically on increasing muscle protein synthesis, which is covered in paragraph 3 and 4 of the introduction (section 1). (2) its specific gastrointestinal properties are referred to in paragraphs 3 and 4 of the discussion (section 4) in order to provide a much lengthier explanation of its role in both muscle protein synthesis and appetite regulation.
- This modification was duly noted and is now in section 3.1, paragraph 1.
- This modification is now added as section 5.
